# Weight Weaving: Parameter Pooling for Data-Free Model Merging

**Levy Chaves**[*]
Recod.ai Lab., Instituto de Computação
Universidade Estadual de Campinas (UNICAMP)
levy.chaves@ic.unicamp.br

**Eduardo Valle**
Intercom[†]
mail@eduardovalle.com

**Sandra Avila**
Recod.ai Lab., Instituto de Computação
Universidade Estadual de Campinas (UNICAMP)
sandra@ic.unicamp.br

**Editors:** Marco Fumero, Clementine Domine, Zorah Lähner, Irene Cannistraci, Bo Zhao, Alex Williams

## Abstract

Model merging provides a cost-effective and data-efficient combination of specialized deep neural networks through parameter integration. This technique leverages expert models across downstream tasks without requiring retraining. Most model merging approaches critically depend on scaling hyper-parameters $\lambda$, which weight each model's contribution globally or individually. Principled approaches for setting scaling factors without accessing any data (data-free) are scarce, often leading researchers to tune $\lambda$ using privileged data from the evaluation set, which is obviously unfeasible in practice. To address this limitation, we introduce Weight Weaving, a plug-and-play technique that pools model weights across $\lambda$ values search space using user-defined pooling functions, such as averaging, random selection, or even existing model merging methods. Our method demonstrates high modularity, imposing minimal constraints on the search space. It operates orthogonally to existing model merging methods and eliminates evaluation data requirements. We validate Weight Weaving across three ViT variants in three experimental setups: vision multi-task learning, vision continual learning, and domain generalization. Our method consistently improves the performance of several model merging methods, achieving average accuracy gains of up to 15.9 percentage points in a data-free setting.

## 1 Introduction

The availability of pre-trained models and public repositories has driven efficient methods for combining and reusing existing models. Researchers have explored various strategies, including model architecture modifications [3, 31], multi-task learning [34], linearization fine-tuning [25, 30], and weight alignment [8, 45]. Model merging represents a promising approach creating unified models by integrating specialized models through efficient parameter-level operations [14, 21, 23, 33]. Task Arithmetic [14] exemplifies this, merging models by combining task vectors — element-wise differences between specialized and pre-trained weights — each scaled by a scaling factor $\lambda \in \mathbb{R}$.

---

[*]Corresponding author. [†] Work developed while at Valeo.ai

Proceedings of the III edition of the Workshop on Unifying Representations in Neural Models (UniReps 2025).

Early works [14, 38] proposed grid search on the validation set to find optimal individual scaling coefficients per task vector. However, as model numbers increase, the search space grows exponentially, requiring $\lambda_i$ per model. To reduce computational overhead, early works [14, 38] simplified this problem by finding a single scaling factor, and subsequent studies [9, 10, 33, 42] adopted the non-standard approach of tuning $\lambda$ *directly on the evaluation dataset*. Other attempts involved test-time training [40, 41], introducing additional hyper-parameters while remaining limited to classification problems. Despite these simplifications, finding scaling factors remains challenging, mainly when validation or evaluation data is unavailable, which is a common scenario in real-world applications.

To address these challenges, we propose **Weight Weaving**, a simple plug-and-play weight pooling method that efficiently marginalizes over arbitrary scaling factors ranges *without requiring any data*. Our method operates on three user-defined inputs: a base model merging function, a search space of scaling factors, and a pooling function to aggregate the resulting parameters. Pooling functions include simple operations like averaging or existing model merging functions. Additionally, our method operates orthogonally to existing model merging approaches, enabling seamless integration with current state-of-the-art (SOTA) methods. We evaluate Weight Weaving across three computer vision applications: multi-task learning, continual learning, and domain generalization, using three ViT variants (ViT-B-32, ViT-B-16, and ViT-L-14). Our method consistently improves SOTA methods in the data-free scenario [10, 38], *i.e.*, when no privileged data is available for finding the scaling factor $\lambda$, with gains up to 15.9 percentage points in average accuracy performance (Figure 1).

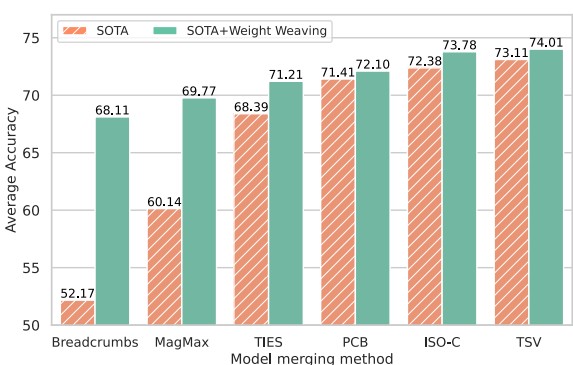

Figure 1: Average accuracy (y-axis) when using privileged data is prohibited. Weight Weaving mostly enhances SOTA merging methods (x-axis) by large margins (Table 2 for details). We consider three ViT model variants across multi-task learning, continual learning, and domain generalization experimental setups. Higher values indicate better performance.

In summary, our contributions are:

- We introduce an efficient weight pooling method collecting parameter-level information across arbitrary user-defined scaling factor search spaces. As far as we know, Weight Weaving is the first pooling framework to bypass privileged data requirements, improving model merger practicality under data-free settings. Additionally, it operates orthogonally to existing techniques, allowing seamless integration with SOTA methods depending on $\lambda$.

- We conduct extensive SOTA evaluations across three experimental setups: vision multi-task learning, vision domain generalization, and vision continual learning. Our method consistently outperforms the standard data-free baseline ($\lambda = 1$), achieving average performance gains up to 15.9 percentage points on SOTA model mergers, as shown in Figure 1.

- We discuss our method's modularity, highlighting practical benefits beyond scaling factor selection challenges. We also outline several promising research avenues and open problems that Weight Weaving introduces, aiming to advance the broader model merging community.

## 2 Related work

Our scope encompasses model merging methods where merged models are fine-tuned from the same pre-trained model, meaning they lie in the same optimization basin [2, 15, 29]. We refer to the survey by Khan et al. [16] for methods that do not assume a shared optimization basin.

**Model merging with the same initialization.** The core idea combines task-specific models into single, versatile models [14, 21, 36, 38] performing all tasks from individual models. Merging operates directly at parameter level, consolidating multiple model weights into a single final model for inference.

Ilharco et al. [14] introduced *task vectors*, the weight differences between fine-tuned models and their pre-trained counterparts. They showed that performing arithmetic on model parameters yields interesting properties, including multi-task learning through weighted combinations of these weight differences into pre-trained models and task negation, which removes task knowledge by negating corresponding weights.

TIES [38] shed light on the magnitude of merged parameters to solve task conflicts, *i.e.*, when parameters benefit one task but not another. They address parameter conflicts through a three-step process: trimming parameters with the highest magnitudes, selecting the appropriate sign, and performing a disjoint merge. Model Breadcrumbs [7] mitigated task conflicts in Task Arithmetic by discarding weight outliers and minor and large perturbations in delta parameters. They find that retaining only mid-range absolute magnitudes improves task compatibility. DARE-TA [42] showed that by randomly setting $p$ percent of delta weights to zero, proportionally rescaling the remaining parameters based on the dropped percentage does not negatively impact merging performance compared to Task Arithmetic. PCB [10] argued that model merging methods should incorporate intra-balancing to adjust importance within tasks and inter-balancing to evaluate similarities across tasks, dropping low-scoring parameters while rescaling remaining ones to improve performance.

MagMax [21] selected parameters with the highest absolute magnitude as an efficient approach for continual learning. TSV [9] measured task interference based on singular vectors interactions from different tasks and uses it to increase merging effectiveness. ISO-C [22] demonstrated how to combine shared and task-specific subspaces for enhanced model merging in multi-task performance. They project individual task-specific subspaces into a shared subspace and aggregate both information.

AdaMerging [41] assumed access to unlabeled test data to optimize merging coefficients automatically by minimizing entropy loss on test data. Representation Surgery [40] also uses test-time optimization to approximate the feature distribution between merged and task-specific models. Both cases demand loading multiple models during training, making the process computationally expensive in time and memory.

**Data-free solutions for finding scaling factors.** MetaGPT [44] proposed a closed-form solution for finding scaling factor coefficients by minimizing average loss of merged and independent models, using a local linearization approach via Taylor expansion and the orthogonality property of task vectors. However, MetaGPT's formulation only considers scaling factors for Task Arithmetic [14], and it does not generalize to other merging approaches and experimental setups beyond multi-task learning. In contrast, our method is a plug-and-play approach, compatible with all existing merging functions. We validate the superiority of our approach and its flexibility to deal with model mergers beyond Task Arithmetic.

## 3 Method

In Section 3.1, we establish the notation and outline the model merging problem. In Section 3.2, we detail the proposed Weight Weaving method, a plug-and-play framework that enhances any model merging methods in a data-free setting. Algorithm 1 shows the steps of our proposed method.

### 3.1 Preliminaries

Here, we briefly describe the parameter-wise merging [9, 10, 14, 21, 38, 39] problem setup. Given a collection of $T$ fine-tuned model weights $\{\theta_1, \theta_2, ..., \theta_T\}$ from the same pre-trained model architecture $\theta_{pre}$, we first compute the set of delta weights $\Delta w = \{\theta_t - \theta_{pre}\}_{t=1}^{T}$ and then merge all delta weights using a merging function $f_{merge}$, such that $\theta_{merged} = f_{merge}(\Delta w, \lambda)$. Here, $f_{merge}$ represents a merging method from the literature (*e.g.*, Section 2), and $\lambda \in \mathbb{R}$ is a scaling factor. We combine $\theta_{merged}$ to the pre-trained model as $\theta_{final} = \theta_{pre} + \theta_{merged}$. For the next sections, we refer to $\theta_{merged}$ as the merged weights and $\theta_{final}$ as the merged model. The model merging problem involves how to combine available weights in $\Delta w$ to output the merged weights $\theta_{merged}$ through $f_{merge}$, without retraining using the training data for each task, and ensuring that $\theta_{final}$ can simultaneously perform tasks $\{1, ..., T\}$.

---

**Algorithm 1:** Weight Weaving

---

**Input:** $\theta_{pre}, \{\theta_t\}_{t=1}^T, f_{merge}, f_{pooling}$
**Output:** Merged model $\theta_{final}$

/* Calculate delta weights, *i.e.*, the element-wise difference between fine-tuned
   and pre-trained models.                                                      */

1  delta_weights $\leftarrow$ {}
2  **for** $t \leftarrow 1$ *to* $T$ **do**
3  $\quad$ $\Delta w_t = \theta_t - \theta_{pre}$
4  $\quad$ delta_weights.insert($\Delta w_t$)

/* Obtain the search space.                                                     */

5  $\lambda_{search} \leftarrow$ get_lambdas_search_space($f_{merge}$)

/* Variable to store the augmented weights.                                     */

6  augmented_weights $\leftarrow$ {}

/* Given an arbitrary merging function $f_{merge}$ and search space $\lambda_{search}$, Weight
   Weaving solves the $\lambda$ search issue.                                    */

7  **for** $\lambda_i$ *in* $\lambda_{search}$ **do**

/* Get the weights to merge according to $f_{merge}$ and $\lambda_i$.            */

8  $\quad$ $\theta_{merge} \leftarrow f_{merge}$(delta_weights, $\lambda_i$)
9  $\quad$ augmented_weights.insert($\theta_{merge}$)

/* Combine deltas and augmented weights to create $A^*$. It has a size of $N \times P$,
   where $N$ is the number of fine-tuned models plus the cardinality of the search
   space, and $P$ is the number of parameters in $\theta_{pre}$.                */

10  $A^* \leftarrow$ { delta_weights $\cup$ augmented_weights}

/* Weight Weaving pools all $\lambda$-scaled parameters from $A^*$ using the pooling
   function $f_{pooling}$.                                                       */

11  $\theta_{pool} \leftarrow f_{pooling}(A^*)$

/* Combine pre-trained model and pooled weights.                               */

12  $\theta_{final} \leftarrow \theta_{pre} + \theta_{pool}$
13  **return** $\theta_{final}$

---

### 3.2 Weight Weaving

The scaling factor $\lambda$ hyper-parameter significantly affects model performance, leading practitioners to often mistakenly set $\lambda$ using data from privileged evaluation sets, which is impractical in real-world situations. In contrast, we propose Weight Weaving, a plug-and-play solution that does not require access to any privileged data, yet still achieves improved merging performance in a data-free experimental setting. Algorithm 1 outlines the complete procedure, consisting of three main steps:

1. **Calculate delta weights.** Given a collection of $T$ fine-tuned model weights $\{\theta_1, \theta_2, \ldots, \theta_T\}$ from the same pre-trained model architecture $\theta_{pre}$, we compute task vectors (delta weights) as $\Delta w = \{\theta_t - \theta_{pre}\}_{t=1}^T$.

2. **Create a set of augmented weights given a user-defined $f_{merge}$.** Our method receives a user-defined merging function $f_{merge}$ and each function has its own search space of scaling factors, denoted by $\lambda_{search}$. Then, for each $\lambda_i \in \lambda_{search}$, we create a new set of augmented weights $A = \{f_{merge}(\Delta w, \lambda_i)\}_{i=1}^{|\lambda_{search}|}$, where $|\lambda_{search}|$ represents the cardinality of the set.

3. **Apply the user-defined pooling function $f_{pooling}$ and merge.** The augmented weight set $A$ includes multiple $\lambda$-scaled parameter variations for merging. We marginalize all those model parameters by applying user-defined pooling function $f_{merging}$ over the set $A^* = \Delta w \cup A$, where $\Delta w$ represents the set of deltas weights to obtain $\theta_{merged} = f_{pooling}(A^*)$. Here, users can adopt a simple pooling function, such as average, random selection, or even any merging $f_{pooling} = f_{merge}$, or adopt any merging function as pooling.

Applying the user-defined pooling function $f_{pooling}$ over the set $A$ can be regarded as a method-centric strategy, since it only considers parameters from the same merging method within specific scaling factors in $\lambda_{search}$. However, we empirically find that incorporating a broader set of parameters

offers benefits in the final performance. Therefore, pooling over the extended collection $A^*$ represents a more collaborative variant, as it combines delta and augmented parameters. Our intuition is that, as the best scaling factor varies depending on the target data and application, pooling over a diverse collection of parameters might be more reliable than making a blind selection.

**Practical considerations.** Our approach is orthogonal to any merging method $f_{merge}$. While we focus on $f_{merge}$ methods requiring tuning $\lambda$, our approach can also be extended to those without $\lambda$ to search any other hyper-parameter. Model merging methods often constrain the scaling factor $\lambda$ to be a real number, but our approach does not impose such limitation. Weight Weavingcan manage any number of variables within a user-defined search space, accommodating not only scalar values, but also categorical variables, probability distributions, or even arbitrary functions as input. The unique constraint is that practitioners must respect the underlying assumptions that make $f_{merge}$ work as expected and adequately handle variables within the search space. This flexibility is essential and highly beneficial to the model merging community, as many prior works [10, 23, 33, 38] require tuning multiple hyper-parameters. Weight Weaving's computation complexity is bounded by the number of elements within the search space and the algorithmic complexity of $f_{merge}$. Our experimental setup covers models up to 422M parameters. However, in the worst case, users can also utilize parallel computing to speed up computations on large-scale experiments involving billions of parameters.

# 4    Experiments

We perform extensive experiments on vision models to demonstrate the effectiveness of our proposed method. In the following, we provide details on our experimental design, including methods, datasets, and evaluation scenarios. For reproducibility, our code is available at `https://github.com/VirtualSpaceman/weight_weaving`.

**Model merging methods.** Our study considers model merging methods that depend on the scaling factor $\lambda$, *i.e.*, the merging is in the form $\theta_{merged} = f_{merge}(\Delta w, \lambda)$, where $f_{merge}$ is a model merging method [10, 14, 38, 42]. We discard methods that require test-time training [40, 41], task-aware inference [13], which limits their use for continual learning and domain generalization, or any post-hoc training [1, 6]. To this point, we consider Task Arithmetic [14], DARE [42], TIES [38], Breadcrumbs [7], MagMax [21], PCB [10], TSV [9], and ISO-C [22].

**Vision multi-task learning.** We follow the setting from [14, 38, 41], which considers ViT-B-32, ViT-B-16, and ViT-L-14, three variants of CLIP [26] models' visual encoders, as the pre-trained models. We fine-tune and evaluate each method on eight image classification tasks: SUN397 [37], Cars [17], RESISC45 [4], EuroSAT [11], SVHN [24], GTSRB [28], MNIST [19], and DTD [5]. Following the fine-tuning phase, we merge the resulting checkpoints from all datasets and evaluate the merged model's performance across the same eight tasks, using accuracy as an evaluation metric.

**Vision continual learning.** We follow the class incremental learning protocol from Marczak et al. [21], using CIFAR100 [18] and ImageNet-R [12] as generic image recognition benchmarks, and CUB200 [35] and StanfordCars [17] for fine-grained classification. We split each dataset into $N$ equal subsets of disjoint classes, where $N \in \{5, 10, 20, 50\}$ for generic benchmarks and $N \in \{5, 10, 20\}$ for fine-grained benchmarks, which contain less data. We train the models using sequential fine-tuning, where training on task $N_i$ starts from the weights $\theta_{n-1}$, previously fine-tuned on tasks $N_1, N_2, ..., N_{n-1}$.

**Vision domain generalization.** We adopt a similar experimental setup to Yang et al. [41] that evaluates the generalization ability of cross-task merged models across different domains. Given a collection of checkpoints to merge, we use a leave-one-out setup, *i.e.*, we hold out one checkpoint, merge the remaining ones, and evaluate the merged model on the dataset corresponding to the excluded checkpoint. We use the same datasets and models as in the vision multi-task learning setup.

**Fine-tuning details.** We follow the training procedure from Ilharco et al. [14] and fine-tune the image encoder with a batch size of 128, a learning rate of $1e$-5 coupled with cosine annealing schedule, and AdamW optimizer with weight decay 0.1 while keeping the text encoder unchanged. We use CLIP's text encoder's final classification layer output and keep it frozen during fine-tuning. This fine-tuning recipe preserves the open-vocabulary nature of the model without compromising accuracy [10, 14, 21].

**Ranges for $\lambda_{search}$.** For most merging methods across continual learning and out-of-distribution setups, we define $\lambda_{search}$ as an equally spaced range from 0.1 to 1.0 with a step size of 0.1. Task Arithmetic, Breadcrumbs, and MagMax consistently follow this standard range across all experimental setups. However, certain methods require expanded search ranges to achieve optimal performance. In vision multi-task learning scenarios, we use broader search ranges of $[0.1, 1.5]$ for TIES, $[0.1, 2.5]$ for PCB, $[0.5, 2.0]$ for TSV, and $[0.1, 2.0]$ for ISO-C, following the search space provided by their respective authors.

## 5 Results

We expect model merging to provide significant benefits to users. By combining knowledge from individual models trained on different tasks, the merged model should achieve competitive test performance across all combined tasks (multi-task learning) or within a single task (continual learning and domain generalization). Here, we present results from three rounds of experiments. First, we compare SOTA merging methods and select the best candidates for further evaluation. Next, we assess our Weight Weaving approach against all competitors in a data-free setting, where privileged data is prohibited. Finally, we investigate the impact of three pooling functions, followed by a discussion of our findings and our method's limitations.

### 5.1 SOTA evaluation

Table 1 shows the average performance of model merging methods across all three experimental setups. For methods that do not consider a data-free setting in their original work, we set $\lambda = 1$. When available, we use the paper's recommended $\lambda$ value obtained with privileged information, denoted by $*$ in Table 1.

Table 1: Average accuracy of merging methods across three vision experimental setups: multi-task learning (MTL), continual learning, and domain generalization. The table presents results for different architectures three ViT variants, and the scaling factor $\lambda$ value adopted for inference. The columns shows the average performance scores across different models and experimental setups, while the last column reports the average performance across all setups. Higher values indicate better performance. Values marked with * are the author's recommended values after tuning on privileged information (evaluation dataset).

| | Setup → | **8 Vision Tasks MTL** | | | **4 Vision Continual Learning** | | | **Vision Domain Generalization** | | | |
|---|---|---|---|---|---|---|---|---|---|---|---|
| | Factor | ViT-B-32 | ViT-B-16 | ViT-L-14 | ViT-B-32 | ViT-B-16 | ViT-L-14 | ViT-B-32 | ViT-B-16 | ViT-L-14 | **Avg** |
| Task Arithmetic | $\lambda = 0.3*$ | 68.37 | 74.86 | 82.91 | 25.18 | 28.61 | 35.10 | 44.70 | 53.54 | 64.57 | 53.10 |
| DARE | $\lambda = 0.3*$ | 68.22 | 74.83 | 82.92 | 25.14 | 28.62 | 35.10 | 44.71 | 53.54 | 64.59 | 53.07 |
| Breadcrumbs | $\lambda = 0.3*$ | 64.22 | 70.21 | 75.67 | 59.40 | 59.98 | 75.57 | 52.33 | 59.82 | 67.59 | 64.97 |
| MagMax | $\lambda = 0.5*$ | 70.49 | 74.23 | 82.02 | 68.53 | 74.64 | 82.38 | 51.76 | 58.51 | 67.10 | 69.96 |
| TIES | $\lambda = 1.0$ | 72.32 | 78.43 | 83.78 | 66.04 | 71.86 | 80.96 | 45.18 | 53.78 | 63.16 | 68.39 |
| PCB | $\lambda = 1.0$ | 75.94 | 80.89 | 86.81 | 67.30 | 73.92 | 81.95 | 51.56 | 58.69 | 65.63 | 71.41 |
| TSV | $\lambda = 1.0$ | 83.60 | 87.23 | 90.47 | 68.33 | 74.79 | 82.84 | 49.72 | 54.47 | 66.54 | 73.11 |
| ISO-C | $\lambda = 1.0$ | 82.69 | 87.03 | 90.60 | 66.64 | 61.74 | 81.44 | 53.20 | 59.17 | 68.95 | 72.38 |

Task Arithmetic and DARE exhibit poor average performance compared to other methods, even when utilizing privileged information, falling more than 10 percentage points behind Breadcrumbs, their next competitor. Particularly, both methods underperformed in the continual learning setup, which was not part of their original evaluation protocol. It suggests that the author's suggested scaling factors may not be transferable to new evaluation settings. Breadcrumbs and MagMax achieved competitive performance to TIES and PCB, but at the cost of requiring privileged information. Based on these results, we exclude Task Arithmetic and DARE from our next round of experiments.

### 5.2 Data-free merging

This is the scenario in which all methods are prohibited from using privileged data because it corresponds to real-world applications. Table 2 compares the average performance of the SOTA methods to our proposed Weight Weavingin a data-free setting, for each experimental setup (columns). Our approach consistently improves average performance across most merging methods, demonstrating significant advantages, particularly in continual learning, and domain generalization. While

Table 2: Data-free average performance per model of the SOTA model merging method across multiple experimental setups. The Avg column shows the average performance across all setups. Higher is better. Our proposed plug-and-play Weight Weaving, which does not require training or extra information, mostly outperforms SOTA methods when privileged data is prohibited. We adopt parameter-wise arithmetic mean as pooling function. We highlight the best value for each row group.

| Setup → | 8 Vision Tasks MTL | | | 4 Vision Continual Learning | | | Vision Domain Generalization | | | |
|---|---|---|---|---|---|---|---|---|---|---|
| Method | ViT-B-32 | ViT-B-16 | ViT-L-14 | ViT-B-32 | ViT-B-16 | ViT-L-14 | ViT-B-32 | ViT-B-16 | ViT-L-14 | Avg ↑ |
| Breadcrumbs | **67.29** | **74.78** | **82.63** | 24.34 | 24.67 | 37.26 | 42.59 | 51.72 | 64.26 | 52.17 |
| +Ours | 66.15 | 71.99 | 77.38 | **66.00** | **70.42** | **80.70** | **52.47** | **60.06** | **67.87** | **68.11 (+15.94)** |
| MagMax | 66.55 | 72.84 | 71.55 | 58.16 | 65.42 | 70.77 | 38.86 | 47.19 | 49.90 | 60.14 |
| +Ours | **69.54** | **74.01** | **82.16** | **67.72** | **73.94** | **81.97** | **51.82** | **58.92** | **67.87** | **69.77 (+9.63)** |
| TIES | 72.32 | **78.43** | 83.78 | 66.04 | 71.86 | 80.96 | 45.18 | 53.78 | 63.16 | 68.39 |
| +Ours | **73.47** | 78.28 | **84.12** | **68.18** | **74.12** | **82.06** | **52.48** | **59.88** | **68.27** | **71.21 (+2.82)** |
| PCB | **75.94** | **80.89** | **86.81** | 67.30 | 73.92 | 81.95 | 51.56 | 58.69 | 65.63 | 71.41 |
| +Ours | 75.55 | 80.77 | 86.43 | **68.31** | **74.24** | **82.02** | **52.88** | **60.29** | **68.42** | **72.10 (+0.69)** |
| TSV | **83.60** | **87.23** | **90.47** | 68.33 | 74.79 | **82.84** | 49.72 | 54.47 | 66.54 | 73.11 |
| +Ours | 80.78 | 86.26 | 89.91 | **68.74** | **75.10** | 82.61 | **53.81** | **60.16** | **68.70** | **74.01 (+0.90)** |
| ISO-C | **82.69** | **87.03** | **90.60** | 66.64 | 61.74 | 81.44 | 53.20 | 59.17 | **68.95** | 72.38 |
| +Ours | 78.33 | 86.03 | 89.12 | **69.43** | **75.60** | **83.07** | **53.71** | **60.22** | 68.51 | **73.78 (+1.40)** |

improvements in vision multi-task learning are less consistent, our method still achieves the best average performance across all setups with gains ranging from 0.90 up to 15.94 percentage points in average. In cases where a baseline occasionally outperforms our approach, the performance gap remains minimal, such as in PCB and ISO-C. Detailed per-dataset performances are available in Appendix A2.

**Discussion.** Weight Weaving consistently enhances domain generalization performance without requiring additional data or optimization. We observe a similar positive trend in continual learning, a setting that has been explored by only a few recent studies [21, 27]. Interestingly, we find that sequential fine-tuning introduces a correlation between the weights of consecutive tasks (Figure 3 in Appendix A3). The sequential process yields task vectors that are not almost orthogonal, contrasting the near-orthogonality in the multi-task learning model merging [14]. We attribute this correlation to the standard practice of initializing the model for task $t$ with the weights from task $t-1$ rather than with the original pre-trained weights. We hypothesize that this increased correlation introduces redundancy among the model's parameters, which may become a significant challenge as the number of continual tasks grows. This observation opens promising new research avenues and can inspire the development of merging techniques specifically tailored for the challenges of continual learning.

## 5.3 Varying the pooling function

Our proposed approach relies on a user-defined pooling function, denoted as $f_{merge}$, which operates over a collection of weights consisting of the delta weights and the augmented weights. Up to this point, we showed the results using a simple average as the default pooling function. Next, we examine how different pooling strategies affect the average performance.

Consider a collection $\mathcal{W} = \{w_1, w_2, ..., w_N\}$ where each weight vector $w_i \in \mathbb{R}^P$ contains $P$ parameters. We denote $w_i^{(p)}$ as the $p$-th parameter of the $i$-th weight vector. We investigate three distinct pooling functions $f_{pooling}$ for weight aggregation:

**Average.** Considers the element-wise arithmetic mean as $f_{avg}(\mathcal{W}) = \frac{1}{N} \sum_{i=1}^{N} w_i$.

**Random uniform selection.** For each parameter index $p \in \{1, ..., P\}$, we independently sample one value from the collection $\{w_1^{(p)}, w_2^{(p)}, \ldots, w_N^{(p)}\}$ with uniform probability $1/N$. We refer this pooling as $f_{rand}$.

**MagMax.** We adopt maximum magnitude parameter selection [21] as an example of using $f_{merge}$ as pooling. For each parameter position $p$, selects the value with maximum absolute magnitude: $\mathcal{W}^{(p)} = w_{i^*}^{(p)}$ where $i^* = \text{argmax}_{i \in 1,...,N} |w_i^{(p)}|$. We refer this pooling as $f_{magmax}$.

Table 3: Average accuracy across three vision experimental setups (multi-task learning, continual learning, and domain generalization), while varying the pooling function. Our results show minimal performance variation between average and random pooling functions. We also tested MagMax, a merging function, as an alternative pooling approach.

| | Pooling function | | |
|---|---|---|---|
| | $f_{avg}$ | $f_{rand}$ | $f_{magmax}$ |
| Breadcrumbs + Ours | 68.11 | 68.11 | 51.93 |
| MagMax + Ours | 69.77 | 69.75 | 60.14 |
| TIES + Ours | 71.21 | 71.21 | 54.56 |
| PCB + Ours | 72.10 | 72.08 | 50.36 |
| TSV + Ours | 74.01 | 73.64 | 55.72 |
| ISO-C + Ours | 73.78 | 73.61 | 64.34 |

Ours results in Table 3 show that average and random uniform selection yields to similar average results. However, by adopting MagMax as pooling function, average results are far worse than previous approaches by a large margin. In this case, merged parameters primarily converge to those with the highest $\lambda$ value, potentially leading to suboptimal performance for some tasks.

**Discussion.** Our exploration encompasses only a subset of the possible pooling functions that practitioners can utilize. A key feature of our approach is its modular design, which allows for seamless integration of any existing model merging method as a pooling function. This flexibility opens promising avenues for future research, particularly in developing pooling methods tailored to specific applications or experimental setups. Such specialized methods could lead to significantly more effective weight combination strategies and improved model performance.

### 5.4 Impact of best scaling factors diversity on performance

To understand why our method demonstrates strong performance in continual learning and domain generalization while showing limited gains in multi-task learning, we examine underlying mechanisms that drive these outcomes. Given that our approach pools parameters from multiple scaling factors, we hypothesize that its effectiveness is related to scenarios where the optimal scaling factor varies across tasks. In other words, our method should provide greater benefits when the best scaling factor for different tasks is distributed across the search space rather than concentrated around a single value.

To validate this hypothesis, we first find the task-specific optimal scaling factors using privileged data. For each task $t$ in our evaluation, we identify the scaling factor $\lambda_t$ that maximizes accuracy on the evaluation dataset of task $t$. Note that this approach differs from the data-free setting, where the absence of task-specific data requires setting $\lambda_t = 1$ for all tasks uniformly. Table 1 shows that our method improves MagMax for all experimental setups, but fails to improve in the multi-task learning setup for ISO-C. To this end, we take these two as examples for our next analysis.

Figure 2 shows the distribution count (y-axis) in which the best scaling factor (x-axis) for the evaluation task across all experimental setups (columns) for the ViT-B-32 model. According to our hypothesis, we expect our method to benefit the latter two column groups more, where the scaling factor distribution is more scattered, and indeed, our approach benefits this case. Again, in a multi-task learning setup, comparing the MagMax distribution to ISO-C's reveals that ISO-C's concentrates around the scaling factor $\lambda = 1$, suggesting that any pooling strategy would not offer significant benefits from this method.

**Discussion.** Best scaling factors vary depending on both the merging method and the target experimental setup. This observation opens important avenues for identifying promising within the search space. Recent study of Li et al. [20] investigated how optimal scaling factors vary depending on task alignment and weight correlations between models to be merged. While their theoretical insights focus on a simplified scenario involving only two models from binary tasks, our work contributes to this discussion by demonstrating that different experimental setups benefit from distinct scaling factor choices. Furthermore, we observe that optimal scaling factors from one experimental setup may not transfer effectively to another, highlighting the dataset-dependent nature of these parameters.

Another promising direction for future work is developing methods for filtering out suboptimal scaling factors within the search space. One approach involves extending the analysis of Li et al. [20], either

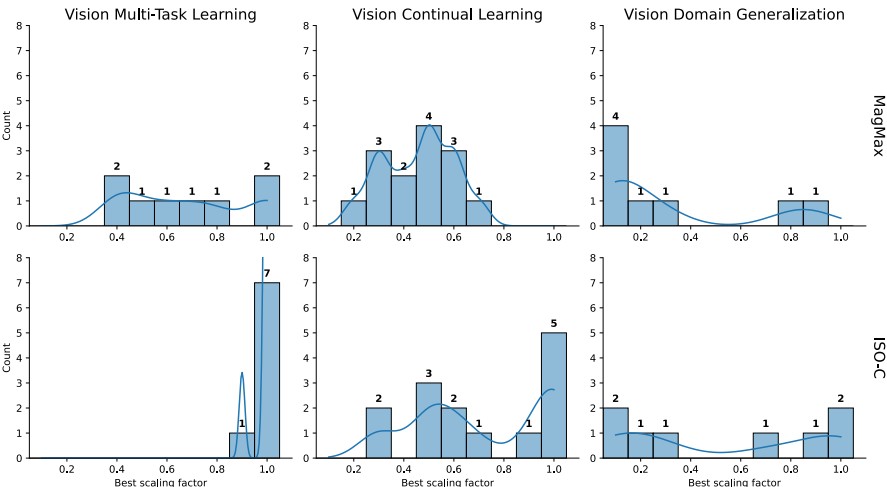

Figure 2: Distribution of optimal scaling factors across experimental setups for ViT-B-32. Each plot is a histogram where the x-axis represents the scaling factor value ($\lambda$), and the y-axis shows the number of times that value yields the best evaluation performance. Each column corresponds to a different experimental setup, while each row represents a model merging method: MagMax (top row) and ISO-C (bottom row). For MagMax, our method consistently improves performance, with optimal scaling factors spread across a range of values. In contrast, ISO-C's scaling factors are concentrated around $\lambda = 1$, in multi-task setting, suggesting that deviations from this value tend to reduce performance. Our method improves on continual learning and domain generalization setups, in which the optimal $\lambda$ values are broadly distributed without a clear central tendency. The blue line in each plot shows the Kernel Density Estimation over the histogram of optimal values.

empirically or theoretically, to more complex scenarios, or by analyzing the interplay between model embeddings and layer-wise weight norms [32, 43]. These advances would also improve our method by allowing it to focus on promising scaling factors rather than exploring the entire search space.

## 5.5 Limitations

The effectiveness of Weight Weaving depends on how the merged model's performance varies across the scaling factors within the search space. For instance, consider a scenario where we define the search space $\lambda_{search}$ in the range [0.1, 1.0], but $f_{merge}$ yields poor performance for $\lambda < 1.0$, with optimal scaling factors concentrated near a single point. In such cases, pooling functions, such as average or random parameter selection, may favor suboptimal scaling factors, which negatively affect the final performance. We observe that such behavior occurs in multi-task learning for ISO-C's, as shown in Figure 2). Mitigating the choice of unstable values within this search range without any privileged information is still an open question.

## 6 Conclusion

This paper addresses a challenge in model merging methods that is the reliance on a scaling factor, $\lambda$, which is often tuned on the evaluated dataset. This dependency on privileged data limits the practicality of many state-of-the-art methods in real-world scenarios where such data is unavailable. To overcome this limitation, we propose Weight Weaving, a data-free, plug-and-play framework that eliminates the need for hyper-parameter tuning. Instead of searching for a single optimal $\lambda$, Weight Weaving pools model parameters across a user-defined search space of scaling factors.

Our method is agnostic to any merging method $f_{merger}$ and requires a user-defined pooling function and a specified search space to function. While our experiments focus on search spaces composed of several $\lambda$, the modular nature of Weight Weaving allows practitioners to search for any number of hyper-parameter or even more complex functions without requiring modifications. We are the first to

introduce the idea of pooling parameters within a user-defined search space to address the issue of searching for $\lambda$ in model merging.

We validate Weight Weaving over three vision experimental setups: multi-task learning, continual learning, and domain generalization. All experiments operate under data-free conditions, reflecting realistic deployment constraints where evaluation data is inaccessible. Across these tasks, our method consistently improves state-of-the-art merging techniques, achieving gains up to 15.9 percentage points in average performance. Additionally, our ablation considers different pooling strategies and their impact on final performance, demonstrating how practitioners can utilize any available model merging method. Finally, we invite the model merging community to explore efficient alternatives to tuning $\lambda$ or to consider additional strategies, rather than relying solely on evaluation data.

**Acknowledgments:** L. Chaves is partially funded by FAPESP (2024/16685-7), CAPES, Becas Santander/UNICAMP – HUB 2022, and Google LARA 2021. S. Avila is also partially funded by FAPESP (2023/12086-9, 2023/12865-8, 2020/09838-0, 2013/08293-7), H.IAAC 01245.003479/2024-10 and CNPq 316489/2023-9, and Google AIR 2022.

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
