# OpenReview forum: "Weight Weaving: Parameter Pooling for Data-Free Model Merging"
_NeurIPS.cc/2025/Workshop/UniReps — UniReps2025_

### Official Review · Reviewer_W4XC · 2025-09-02

**Confidence:** 4

**Review:**

## Summary

Existing merging methods mostly rely on validation data for hyperparameter tuning, specifically the merging coefficient lambda. In cases where the user has no access to validation data, there is no principled way (as far as I know) of picking lambda. This paper thus introduces Weight Weaving, a plug-and-play framework for data-free model merging that bypasses lambda tuning. It first computes the delta parameters (also AKA task vectors) from individually finetuned checkpoints, then applies a user-specified merging function to fuse these delta parameters across all possible lambda values in a specified range, obtaining a set of multitask deltas derived all from the same merging function but with different magnitudes scaled by the lambdas. We unify these new multitask deltas with the original deltas and apply a pooling function to obtain a final multitask delta to be summed to the base model. No data is involved throughout.


## Strengths

### Novelty and Practicality:
- Tackles an important limitation in model merging, i.e., the reliance on validation data.
- Introduces a pooling framework applicable across merging methods.
- As a proof of concept, it has a lot of possible future extensions to explore.

### Modularity:
- Works orthogonally with existing model merging approaches, making it easy to integrate.
- Flexible with respect to both search spaces (scalars, distributions, functions) and pooling functions.

### Experimental Validation:

 - Thorough evaluation across three setups (multitask, continual, domain generalization).
 - Demonstrates clear improvements in continual learning and domain generalization.

## Weaknesses
- Weak applicability in multitask merging: model merging is currently predominantly applied for multitask purposes, and the method shows no clear gains in this setting, and often even a deterioration.  This significantly weakens the usefulness of the method.
- Lambda is a single scalar value applied for all tasks; there's no task-specific lambda, as in E.g., AdaMerging.
- The concept of the lambda search range is vague and arbitrary. For a specific merging function, what is a principled way to infer its search range? The range itself seems another hyperparameter. Why is [0.1, 1] better than [0.1, 1.5], for example?
- Lack of theoretical grounding: the theoretical reason why the method works is not sufficiently discussed.
- For very large models, applying merging and pooling that many times is costly.
- Although a high degree of freedom, there's a lot of homework for the user to do, and non-experts might have a hard time picking good merging and pooling functions.


## Questions for the Authors
1. How sensitive are the results to the step size and range of the lambda search space?
2. Why does the method work, theoretically, or at least, intuitively?
3. Since the augmented set A contains the same multitask delta scaled with different lambdas, if you use vanilla average as the pooling function, isn't merging original deltas with the augmented set A essentially equivalent to merging the original deltas with the mean delta in A? If so, the mean delta in A could have been computed directly as f_merge(Δw, λ_mean), where λ_mean = mean{λ1, λ2, ...}. Therefore, the augmented set doesn't even have to be computed.
4. Is this method really efficient? The actual efficiency heavily depends on the merging function, pooling function, search step size, and search range.


## Suggestions
1. Provide more theoretical and intuitive grounding for the method.
2. Compare the computational workload, which for large models might be prohibitive.
3. Provide some guidance for the users on how to reasonably choose the merging and pooling functions, as well as the search step size and range.
4. In line 131, I wouldn't say "mistakenly" using privileged data. Neither finding a suboptimal lambda nor using available privileged data is a mistake. If you believe it is a mistake, please elaborate further.

**Score:**

2

**Topic Fit:**

3

---

### Official Review · Reviewer_7XdV · 2025-09-05
**Review of Weight Weaving: Parameter Pooling for Data-Free Model Merging**

**Confidence:** 4

**Review:**

### **Summary**
This paper introduces Weight Weaving, a data-free, plug-and-play framework for model merging that pools parameters across a user-defined scaling factor search space. It operates orthogonally to existing SOTA merging methods and consistently improves performance in continual learning, domain generalization, and multi-task vision benchmarks.

### **Strengths**
- The technical contribution is innovative, clear and well explained.
- Data-free and modular: Can be applied on top of existing SOTA model merging techniques without access to evaluation data.
- Particularly effective when optimal scaling factors vary across tasks, common in continual learning and domain generalization.
- Extensible research directions: Opens avenues for optimizing pooling functions, for instance through data-free entropy criteria or meta learning-based methods.

### **Weaknesses**
- Limited evaluation domains: Experiments are restricted to visual tasks (ViTs); generalization to NLP, multimodal, or other domains is not demonstrated, although in principle the approach should extend (possible direction for future work).
- Section 5.4 shows that pooling yields poor performance when the optimal scaling factors are concentrated near a single point. This observation is well supported by the empirical results and plots. Since it is central to the paper, it would be helpful to include a few lines providing a more formal explanation of why pooling across similar scaling factor values is ineffective.

### **Additional Comments**
- Extensive empirical evaluation and comparison with several SOTA vision model merging methods is appreciated.
- Appendix A3 provides insightful discussion on weight correlations in sequential vs. individual fine-tuning.
- **Minor fix:** citations [9] and [10] appear to reference the same paper, likely due to a bibliography error.

**Score:**

4

**Topic Fit:**

3

---

### Official Review · Reviewer_m6qK · 2025-09-16
**Plug-and-Play Expert Parameter Pooling**

**Confidence:** 5

**Review:**

## Evaluation

The paper is of high quality, with thorough experiments, strong baselines, and ablations that clarify when the method works best. Clarity is generally good, though some terms (e.g., “selfish/unselfish pooling”) are informal. The contribution is original, reframing $\omega$-search as parameter pooling, a simple but elegant idea that generalizes across merging methods. The significance is practical rather than theoretical: it eliminates a major barrier in real-world merging (need for evaluation data) and opens new research directions, though scope is limited to vision models and efficiency is not deeply studied. Finally, further theoretical analysis on what the pooling of parameters do, in terms of filtering task vector contributions, has yet to be done.

## Pros

1. A data-free merging method: no dependency on “privileged” evaluation data
2. Plug-and-play nature: can be inserted on top of other merging methods, such as task arithmetics, TIES, Breadcrumbs, etc.
3. Shows competitive SOTA performance improvements, especially in domain generalization tasks.
4. Framework allows plugging different pooling operations (flexibility).
5. Good CL-related insights: from task correlations to smart $\omega$ selection.

## Cons
1. Heavy dependance on optimal $\omega$ values
2. Lack of analysis on pooling granularity: pooling is always applied parameter-wise across the whole model, i.e.e, no per-layer, per-block strategies.
3. Pooling function choice matters: careful hyper-parameter selection is crucial
4. The treatment of selfish/un-selfish pooling seems a bit informal
5. There are some typos, especially in the method section

**Score:**

4

**Topic Fit:**

3

---

### Official Review · Reviewer_rvUv · 2025-09-16

**Confidence:** 4

**Review:**

This paper addresses a critical practical limitation in model merging research: the reliance on privileged evaluation data for tuning scaling factors in merging methods. The authors propose Weight Weaving, a data-free technique that pools model weights across a user-defined search space of scaling factors using simple pooling functions (e.g., averaging, random selection). The method operates orthogonally to existing merging approaches and demonstrates consistent improvements across three vision experimental setups: multi-task learning, continual learning, and domain generalization.

# Strengths
- The paper tackles a real-world problem in model merging where privileged data is unavailable, making it highly relevant for practical deployment scenarios.
-  Weight Weaving is remarkably straightforward to implement requiring minimal computational overhead while achieving significant performance gains.
-  The authors rigorously evaluate their method across three distinct experimental setups (multi-task learning, continual learning, domain generalization) using three ViT variants, demonstrating consistent improvements over SOTA methods.

# Weaknesses
- The paper doesn't provide theoretical justification for why Weight Weaving works so well in continual learning and domain generalization but shows limited gains in multi-task learning for some methods (e.g., ISO-C)

- The paper doesn't explore how Weight Weaving differs from DiTASK [1]. DiTASK generalizes parameter-efficient fine-tuning methods and specifically for MTL offers weight matrix modulation using a diffeomorphic transformation on singular values. Can Weight Weaving learn better parameters for MTL than DiTASK and how does it compare wrt MTL performance?


[1] Krishna Sri Ipsit Mantri, Carola-Bibiane Schönlieb, Bruno Ribeiro, Chaim Baskin, Moshe Eliasof; Proceedings of the IEEE/CVF Conference on Computer Vision and Pattern Recognition (CVPR), 2025, pp. 25218-25229

**Score:**

4

**Topic Fit:**

3

---

### Decision · Program_Chairs · 2025-09-23

Accept (oral)